# Study of Tribological Properties of Fullerenol and Nanodiamonds as Additives in Water-Based Lubricants for Amorphous Carbon (a-C) Coatings

**DOI:** 10.3390/nano12010139

**Published:** 2021-12-31

**Authors:** Shuqing Chen, Qi Ding, Yan Gu, Xin Quan, Ying Ma, Yulong Jia, Hongmei Xie, Jinzhu Tang

**Affiliations:** 1College of Materials Science and Engineering, Yangtze Normal University, Chongqing 408100, China; 20170054@yznu.edu.cn (Y.G.); quanxinjianshe@163.com (X.Q.); yma2017@126.com (Y.M.); yljia2017@126.com (Y.J.); xiehongmei@yznu.cn (H.X.); 2State Key Laboratory of Solid Lubrication, Lanzhou Institute of Chemical Physics, Chinese Academy of Sciences, Lanzhou 730000, China; dingqi@licp.cas.cn

**Keywords:** fullerenol, nanodiamonds, water-based lubricant additive, a-C, nano-bearing effects

## Abstract

The tribological performances of fullerenol and nanodiamonds (NDs) as additives in water-based lubricants for amorphous carbon (a-C) coatings are investigated to avoid disadvantage factors, such as chemical reactions and deformation of particles. The effects of size and additive amount on tribological properties of nanoparticles are studied by rigid nanoparticles within the dot size range. The results show that owing to its small particle size (1–2 nm), fullerenol cannot prevent direct contact of the friction pair at low concentration conditions. Only when the quantity of fullerenol increased to support the asperity contact loads in sufficient concentration did nano-bearings perform well in anti-friction and anti-wear effects. Unlike fullerenol, nanodiamond particles with a diameter of about 5–10 nm show friction-reducing effect based on the nano-bearing effects at ultra-low concentration (0.01 wt.%), whereas particles at higher concentration block the rolling movement, hence increasing the coefficient of friction (COF) and wear. As a result of the effect of difference in size, fullerenol provides a better overall lubrication, but it is hard to reach a friction coefficient as low as NDs even under the optimal conditions.

## 1. Introduction

With the increasing of the energy resources consumption and industrial pollution, the issues of energy and environmental protection increasingly become the focus of research in many fields including tribology. Many eco-friendly lubricants and environmentally friendly additives have been researched and developed to replace conventional oil and grease lubricants for alleviating energy crisis and solving environmental problems. As the most environmentally friendly, abundant, and renewable resource on earth, water has its own unique advantages in energy saving and environmental protection. Thus, water-based lubricants have been investigated widely and also applied as cutting, machining, or drilling fluids in metal forming operations and oil extraction. However, water is a medium with very low viscosity, showing inadequate lubricity and having a corrosive effect on many metals. These defects strictly limit the further application of water-based lubricants in the industry [1]. Accordingly, for the past decades, a growing number of studies have been focused on the promotion of water-based lubrication and corrosion resistance, suggesting that the tribological properties of friction pairs as well as the additive effect can strongly affect the tribological performances of water-based lubrication [2,3,4,5].

Owing to its exceptional properties including high hardness, low friction coefficient and good chemical inertness, amorphous carbon (a-C) coatings have excellent tribological properties with low coefficients of friction and wear rates in water, and their deposition has been recognized as a highly effective approach to improve the corrosion resistance of tribopairs under water-lubricated conditions [6,7]. However, it was found that a-C exhibits low reactivity with many chemical additives because of their chemical inertness. Recently, many studies have demonstrated graphene and its derivatives have a significant positive effect on improving the tribological behavior of a-C contacts [8,9,10].

On the other hand, carbon nanomaterials can significantly enhance the tribological properties of water-based lubricants with excellent properties such as self-lubrication, good water solubility, and stability for potential application in water-based lubrication additives [11,12,13,14,15]. For example, Jiao et al. investigated the water-based lubricating property of bioinspired surface functionalization of nanodiamonds (ND-MPC), and they found that due to the rolling effect of the nanoparticles, both the wear and coefficient of friction are further decreased by adding ND-MPC to water-based lubricants [13]. Elomaa et al. investigated the lubrication property of water by adding graphene oxide (GO) nanosheets, and they found that the wear volume loss of tribopairs was significantly reduced by GO [12].

With the develop of the research, more and more kinds of carbon nanomaterials are used as water-based lubrication additive, and their tribological properties and lubrication mechanism are widely investigated. Recently, a new type of carbon material, carbon dots (CDs), have attracted considerable interest in the field of tribology, especially in water-based lubrication. Compared with the other carbon nanomaterials, CDs have smaller sizes and better solubility in water which can solve the embedded stability problem between micro-bulges of rubbing interfaces. Therefore, CDs show excellent friction reduction and anti-wear performances in water as lubricant additives. For example, Liu et al. reported that the CDs-IL (ionic liquid) can increase the load-carrying capability and enhance the lubricity of water for steel contacts [16]. Xiao et al. found that sulfur-doped CDs show good anti-wear and anti-friction properties as water-based lubricant additives for Si_3_N_4_-vs-steel and Si_3_N_4_-vs-Si_3_N_4_ contacts [17]. Hu et al. have researched the enhanced tribological properties and inhibition effect of CDs in water-based lubricants for 316 stainless steel [11]. More recently, Tang et al. used CDs as water-based lubrication additives for a-C. The results showed that CDs exhibited remarkable friction and wear reduction effects when the content of CDs was only 0.1 wt.% [18].

The excellent tribological properties of CDs are usually attributed to the synergistic action of multiple lubrication mechanisms including the formation of tribofilm, the rolling effect, interlayer shear sliding effect, and nano-filling effects. It is very hard to accurately describe the tribological performance of CDs by using a single mechanism. Therefore, the correlative researchers are often confused about the positive approach and key development direction to further improve the performance of CDs. Furthermore, even though CDs have good solubility in water, they are still instable during friction. As reported in our previous article, the force of friction deteriorated the solubility of CDs particles and CDs subsequent in situ aggregated by forming larger particles [18]. In fact, the loss of CDs will cause degradation in performance in long-term use, particularly at a considerably low concentration (0.1 wt.%). In summary, it is very important to strengthen the basic research on the mechanism and development of long-acting water-based lubricant.

As an important lubrication mechanism of nano-additives, the effect of nano-bearing as the reason for the improved tribological performance is hard to measure accurately, partly due to hardly monitor changes in the morphology and motion of nanoparticles directly in real time [9,11,18,19,20]. In fact, the deformation, exfoliation, and sedimentation of nanoparticles are adverse to the rolling motion of nanoparticles. Additionally, the general studies of water-based nanolubricants tend to shift towards smaller particle size which can improve the stability, solubility, and tribological properties in water because of its small size effects and interaction of surface groups. As spherical nanoclusters at molecular level, fullerenol and its derivatives with good water solubility and stability show great application potential as water-based lubricant additives because of its unique structure and excellent physical properties [21,22,23,24,25,26,27]. To the best of our knowledge, however, research about the tribological properties of water-soluble fullerenol are very few at present and lack systematical research results. Particularly in regard to the friction pair of a-C having excellent tribological properties in water, there is no literature that reports on this subject.

In this study, two types of rigid dot sizes carbon particles (i.e., fullerenol and NDs) are selected as lubricant additives for water-lubricated a-C/a-C contacts to avoid the disadvantage factor of nanoparticles described above. The tribological performance and possible boundary lubricating mechanisms are explored from particle size and concentration. It is very interesting that the lubrication effect of fullerenol increases with concentration, which is contrary to that of NDs.

## 2. Materials and Methods

### 2.1. Materials

In this research, fullerenol was synthesized by using C60 of mass fraction purity 99.9% as precursor. The C60 and nanodiamond (97.0%) with a particle size 5–10 nm were purchased from Nanjing XFNANO Materials Tech Co., Ltd., Nanjing City, China. Additionally, 40% tetrabutylammonium hydroxide (TBAH) was purchased from Aladdin Reagent Co., Ltd., Los Angeles, California, USA. Benzene (99.8%), methanol (99.7%), and sodium hydroxide (98%) were purchased from Sinopharm Chemical Reagent Co., Ltd., Shangai, China. The nonhydrogenated a-C film without doping elements was deposited on GCr15 steel discs and balls with a diameter of 6 mm by using UDP650 unbalanced DC magnetron sputtering deposition system (Teer Coatings Ltd., Worcestershire, UK), and the basic properties of analyzed a-C film are listed in Table 1.

### 2.2. Fullerenol Preparation

The C60(OH)_22–26_ was synthesized by the most simple and repeatable method as presented by Semenov K N [23] and Bityutskii N P [28]. Fullerene, 7.5 mg in weight, is dispersed in 1000 mL of benzene and stirred at room temperature during 20 h. Next, the solution is filtered through a Schott filters (porosity factor 10). Then 20% tetrabutylammonium hydroxide solution (1.5–2.0 mL) and 50 wt.% sodium hydroxide aqueous is slowly added to the C60 solution in benzene up to decoloration of the benzene solution. It is thereafter vacuum distillation of the benzene; a further 100 mL distilled water is added to the reaction mixture. The reaction mixture is then stirred for 20 h. After the reaction is completed, 200 mL water is added to the reaction mixture. The mixture is filtered through the Schott filter (porosity factor 10), and the solution is evaporated using a rotary evaporator up to 50 mL. A quantity of methanol (500 mL) is added for precipitating fullerenol from the aqueous solution (the reprecipitation procedure is repeated thrice). The precipitate is separated from the solution and washed with methanol neutral value of pH = 7, then vacuum-dried at 40 °C and 13.3 Pa. Finally, the fullerenol reddish brown powder is obtained.

### 2.3. Tribological Evaluation

As carbon additives have excellent tribological properties, particularly hard particles with low addition, the nano additives are first added at a series of low concentrations of 0.01, 0.05, and 0.1 wt.% to evaluate the differences of tribological properties between fullerenol and nanodiamond and point out the main influence factors clearly. Both the fullerenol/water nanofluids and nanodimond/water nanofluids show excellent dispersion stability, particularly the prepared fullerenol with good water solubility which remains stable in water for more than six months (see Appendix A). Then the influencing factors are further studied. All tribological tests are carried out by a reciprocating ball-on-disk UMT-2 tribometer at room temperature. The boundary lubrication friction tests are carried out in ambient air (25% relative humidity) under 10 N applied load with a frequency of 5 Hz (relative contact velocity of 0.05 m/s) for 30 min. The initial Hertzian contact stress is about 0.93 GPa. All experiments are repeated at least 3 times to yield statistically relevant results.

### 2.4. Characterizations

The composition, morphology, and structure of fullerenol and NDs are measured by transmission electron microscope (TEM, Tecnai G2), Fourier transform infrared spectrometer (FTIR, Nicolet iS20), and XPS (ESCALAB 250Xi, ThermoFisher Scientific, Waltham, MA, USA) spectroscopy. After the tribological tests, the worn a-C surfaces are ultrasonically cleaned in alcohol and acetone each for 10 min. The JEM-5600LV scanning electron micro-scope (SEM; JEOL) and 3D optical profilometers (3D Profilm, Filmetrics, San Diego, CA, USA) are employed to observe the morphology of worn surfaces and measure the wear volume. The formula (K = V/(FS), V is the wear volume, F is the applied load, and S is the total sliding distance) proposed by Archard and Hirst is used to calculate the specific wear rate K. The structural changes of a-C surface during friction tests are evaluated by the Raman spectroscopy (Lab JY-HR800, Horiba, Tokyo, Japan, l:532 nm) spectroscopy and XPS (ESCALAB 250Xi, ThermoFisher Scientific) spectroscopy.

## 3. Results

### 3.1. Fullerenol and Nanodiamond Characterizations

As shown in Figure 1a, the fullerenols are circular-like particles with average diameters of 1.8 nm and well dispersed without aggregation. The lattice distance (0.213 nm) corresponding to d-spacing of fullerenol lattice fringes is clearly observed in high-resolution TEM (HRTEM) images (Figure 1a) [29,30]. The similar sphere-shaped morphology of NDs with average diameters of 6.8 nm is shown in Figure 1b. The electron diffraction pattern (in the bottom right-hand corner) indicates that the crystal structure of these particles is that of the substantially ordered cubic diamond, but the crystallite is not perfect because of the widened diffraction pattern. The HRTEM image (in the right-hand upper corner) also shows that the NDs are diamond single crystals only with defects at the particle surface. The statistical size distribution of fullerenols and NDs are measured from the SEM photographs using an image analysis software (nano measure 1.2). As can be seen from Figure 1c, all fullerenols have ultra-small diameters in the very narrow 1–3 nm range. The NDs have diameters of 4–10 nm and most particle diameters are larger than 5 nm.

The FT-IR spectra of fullerenol and NDs are shown in Figure 2. As invariably reported as the diagnostic absorptions of various fullerenols, there is a broad band at 3680–3100 cm^−1^ (O-H stretching), and three absorption characteristic bands at 1600 cm^−1^ (C = C stretching), 1380 cm^−1^ (C-O-H stretching), and 1080 cm^−1^ (C-O stretching) [31,32]. The FT-IR spectrum reflects that the major functional groups of fullerenol are hydroxyl groups. The IR spectrum of NDs has two absorption peaks at 3435 cm^−1^ (O-H stretching) and 1637 cm^−1^ (C=O stretching) presenting carboxyl and hydroxyl groups on the surface. Those oxygenated functional groups cause surface defects (see Figure 1b) and benefit the dispersion and stabilization of NDs in water. In fact, both fullerenols and NDs could be well dispersed in water and can remain suspended in water for 12h (see Appendix A). The chemical composition of fullerenol and NDs are further confirmed by XPS. As can be seen from Figure 2b, the prepared fullerenol shows three main C1s peaks centered at 284.8, 286.4, and 288.3 eV, which are attributed to nonoxygenated carbon, monooxygenated carbon, and hemiketal carbon, respectively [33]. Four bands assigned for sp2 C-C bonds, sp3 C-C bonds, carbon bonded to oxygen as C-O, and C−H bonds can be identified in the spectrum of NDs, just as in Figure 2c [34,35]. In addition, the binding energies for fullerenol and NDs are obvious different because of the changes of the chemical state of carbon.

### 3.2. Tribological Results

Figure 3a shows the friction coefficients of fullerenol dispersion and NDs dispersion for a-C/a-C contact. It can be observed that both fullerenol and NDs as additive improve friction reducing effect of pure water lubrication, but their variation tendency with concentration are different. As the adding amount increases, the friction reducing effect of fullerenol increases continuously. On the contrary, for NDs, the optimal effect of antifriction effect at the minimum adding amount, and show decreasing tendency varies with concentration increases. When the additive concentration increases to 0.1 wt.%, the fullerenol dispersion and NDs dispersion show similar low friction values of 0.04. Friction is reduced by about 30%, compared with that result from the use of deionized water as a lubricant.

Figure 3b shows the ware rate of a-C contacts lubricated with fullerenol dispersion and NDs dispersion. It shows that the antiwear effect of fullerenol becomes significant only when the adding amount increase up to 0.1 wt.%. Compared with deionized water, the wear rate is reduced to 1.9 × 10^−8^ mm^3^ (N·m)^−1^, which is lower by 42%. In contrast, NDs have a slightly antiwear effect only at the lowest concentration, but significantly deteriorate the antiwear properties of water when the additive concentration increases up to 0.05 wt.%. In addition, the deterioration further proceeds with the concentration further increasing. As shown in Figure 3b, the wear rate of a-C under the lubrication by 0.1 wt.% NDs dispersion increases rapidly to 8.3 × 10^−8^ mm^3^ (N·m)^−1^ which is about 1.5 times of the result from the use of deionized water as a lubricant. The different tribological performances between fullerenol and NDs according to the adding concentration may originate from different contact state with particles because of the size effect as discussed in later sections.

To further evaluate the tribological performance of fullerenol and NDs in water, the tribotest are conducted under different normal load conditions, as shown in Figure 4. It can be seen that fullerenol have almost same wear rates under all conditions, showing good anti-wear properties. However, the anti-friction properties of fullerenol are dependent upon the normal load. Fullerenol show good anti-friction properties when the normal load is lower than 10 N. However, the anti-friction effects of fullerenol become less as the normal load increased to 15 N. On the contrary, the effects of NDs in anti-wear are observed only in low load condition. When the normal load becomes greater than 10 N, the adverse effects of NDs are observed where the wear is significantly increased by 11%. NDs show good friction-reducing properties under all conditions; however, the anti-friction effects of NDs decreased with increasing normal load. In addition, fullerenol dispersions exhibit much lower wear but higher friction compared with NDs dispersions under the same condition. In addition, the tribological performance of dispersions with the same concentration is compared in Appendix A. Just as in tribological results discussed above (Figure 3), at low concentrations of 0.01 wt.%, fullerenol have less effects on tribological performance of a-C, and at the concentration of 0.1 wt.%, NDs significantly worsen the anti-wear properties of a-C under water lubrication.

Considering the result that high content of fullerenol is beneficial for acquiring the excellent lubrication performance, the tribological performances of fullerenol at higher concentrations are further studied. As can be seen from Figure 5, in deionized water, the COF gradually increases up to high level of COF (0.08) after a long running-in period. The running-in time is obviously shortened and the steady-state friction coefficient is significantly reduced by adding fullerenol to the water. It should be noted that there is no obvious change in the friction coefficients and wear rates with further increase in the fullerenol additive content to more than 0.1 wt.%. In comparison, 0.1 wt.% fullerenol dispersion exhibits more stable friction coefficient curves.

### 3.3. Characterizations of Worn a-C Surfaces

Figure 6 shows the morphology and cross-sectional profile of the wear track on the a-C surface after the tribological tests under the applied load of 10 N. It shows that in deionized water, abrasive wear took place in a-C contacts indicated by the obvious deep grooves in the wear tracks region (Figure 6a). Introducing 0.1 wt.% fullerenol into water could remarkably alleviate the abrasive wear which is evidenced by the smooth worn surface. The maximum wear depth is 43 nm which is considerably lower than that when pure water is used as a lubricant (98 nm). For NDs, quite different effects are observed. NDs could slightly alleviate the abrasive wear at very low concentrations (0.01 wt.%). When the amount of adding NDs is slightly increased up to 0.1 wt.%, the abrasive damage, however, becomes particularly severe. As can be seen from Figure 6d, the U-shaped wear grooves are formed on the worn surface with a significantly increased wear depth. The maximum wear depth is about 224 nm, which is more than twice as the result from the use of deionized water as a lubricant.

The microscopic images of wear scar on the slider balls are shown in Figure 7. It is seen that compared with that of a-C lubricated by deionized water, the surface of wear scar is smoother and the diameter of wear scar is smaller under the lubrication of 0.1 wt.% fullerenol dispersion. This result suggests that introducing 0.1 wt.% fullerenol into water could alleviate the abrasive wear. NDs show quite different effects. At the low additive amount of 0.01 wt.%, NDs had little effect on the appearance and diameter of wear scars but enlarge the diameter of wear scar and increase the number of deep grooves on wear scar at the additive amount of 0.1 wt.%.

The morphology of worn surface is further investigated by SEM, and the results are shown in Figure 8. It can be seen that under the lubrication of water, the main wear mechanism of a-C is slightly abrasive wear because there are multiple parallel grooves and without typical damage morphologies of tearing crater caused by severe adhesion. As shown in the high-magnification SEM image (Figure 8a), the worn a-C surface still maintains its original appearance with tightly packed and rounded micro-regions. The worn surface lubricated with water containing 0.1 wt.% fullerenol is considerably smooth and without visible grooves indicating that fullerenol could remarkably alleviate the abrasive wear. However, for NDs, the worn mechanism of a-C is highly sensitive to addition concentrations. As can be seen from Figure 8c, the abrasive wear behaviors could be slightly alleviated by 0.01 wt.% NDs dispersion which is evidenced by the smooth worn surface and shallower grooves on the wear track.

In contrast, NDs significantly deteriorate abrasive wear performance of a-C when the additive amount slightly increases up to 0.1 wt.%. The high-magnification SEM image in Figure 8d shows that in 0.1 wt.% NDs dispersion, some micro cracks are observed in the wear track. The results relate to the fact that the load-bearing capability of a-C films are reduced as a-C films become thinner which will eventually result in the formation of the crack array. It should be noted that for fullerenol and NDs, there are no additive particles attached to the worn a-C surface. Therefore, the mechanisms of self-healing effect and tribolayer formation based on the adhesion of nanoparticles are not the main determinant of the tribological properties of fullerenol and NDs.

The chemical composition changes on a-C surfaces after tribological tests are investigated via Raman spectroscopy and XPS. Figure 9 shows the Raman spectra obtained from the as-deposited a-C and the worn a-C surface. All spectra of the worn a-C surfaces are considerably similar to that of as-deposited a-C showing the similar broad peak between 1100 and 1750 cm^−1^. The results suggest that the a-C coating preserved the amorphous structure during friction. The spectra are further deconvoluted by the BWF + Lorentzian pair into two main Raman bands, with one locates around 1550 cm^−1^ (G band), and the other around 1360 cm^−1^(D band). The fitting results are corresponding inserted in Figure 9. It can be seen that there is no obvious change in G peak position (Gmax), and the I_D_/I_G_ ratio do not change, suggesting that the a-C film maintain the native structure during test. This result is also confirmed by XPS as no obvious changes in the C1s spectrums obtained from the a-C surfaces are found before and after tribological tests (see Figure 10). In addition, the worn surfaces of a-C contain three types of C atoms: centered at 284.8 eV (C = C/C-C), 286.1 eV (C-OH), and 287.5 eV (C-O-C), which are the common chemical composition of amorphous carbon film [36,37,38]. The characteristic peaks of as-prepared fullerenol and NDs are not found in the corresponding worn a-C surfaces, indicating there are no three body particles deposited on worn a-C surfaces.

## 4. Discussion

As presented above, in the aspect of lubrication effect for a-C contacts, the fullerenol is better than NDs because the former has excellent effects in both friction and wear; furthermore, it is applicable over a wider concentration range. As the results of the worn surface morphology analysis, both fullerenol and NDs act as the third body particles between two rubbing surfaces, and most of wear debris and nanoparticles are washed away from the worn surfaces by the shear force and liquid during the sliding process. Additionally, the existence of nanoparticles in water does not promote the graphitization of a-C which is the main tribological mechanisms of a-C (see Figure 8) and they do not deposit on the worn surface during sliding tests. Therefore, we assume that the plausible lubricating mechanisms of fullerenol and NDs is the nano-bearing effect.

The different lubricating performances of fullerenol and NDs as lubricant additives in water are related to the different contact state with particles because of the different nano-sizes. As both fullerenol and NDs are known to be rigid nanosized carbon particles, the yield stress and tensile strength of the two materials are higher than the contact stress. It can be assumed that fullerenol and NDs are able to retain their basic original morphology during frication. Spherical nanoparticles enter the interface of tribo-pairs and behave as nano-rolling and nano-bearing separating the two mating surfaces and converting sliding friction into rolling friction, thus significantly reducing friction coefficient and wear. Therefore, the reduction of the asperity–asperity contact and the ratio between the rolling friction term and the sliding friction term are the key parameters for controlling the effect of nanoparticles.

As can be seen from Figure 11d, nano-peaks and nano-valleys with 4–15 nm heights/depths are present on the a-C surfaces because of the dense packing of carbon atoms with densely packed spherical grains. As the particle size of fullerenol is only 1–2 nm, compared with fullerenol particles, the surface roughness of a-C is highly rough. Thus, fullerenol particles do not prevent frictional interfaces from direct contact and cannot reduce plastic deformation of a-C when insufficient particles presented in contact interface (see Figure 11a). Therefore, as the tribological results show (see Figure 3), fullerenol has less effect on the lubrication properties of water at low concentration because the nano bearing effect is unlikely to take place. With the increase of adding amount of fullerenol, more fullerenol stays in the contact area and the area of fullerenol surface coverages becomes higher (see Figure 11b). The applied load stress of fullerenol decreases at high fullerenol surface coverages, which is beneficial for separating the two mating surfaces and rolling friction, hence decreases the friction and wear.

Unlike fullerenol, NDs with particle size of 5–10 nm are almost the same as the peak-to-valley roughness of a-C which can effectively support the asperity contact loads at very low concentration, and thus provides a very low coefficient of friction (see Figure 11c). Increasing the concentration of NDs, however, will block the rolling movement of particles and limit the motion of the neighboring particles, leading to the abrasive behavior, and hence increasing the COF and wear.

It is noteworthy that even though fullerenol provides a better overall lubrication, it is hard for fullerenol to reach a friction coefficient as low as NDs. This is because the reduction in real area of contact with NDs is more than fullerenol. For boundary lubrication, the frictional force is mainly caused by the relative sliding between two solid contact surfaces and viscous stress of liquid lubricants. The viscous stress of liquid lubricants is usually small to be ignored. The solid-solid contact area can be divided into direct asperity-asperity contacts and particle lubricating area, and hence the friction force consists of the corresponding two parts. Assuming that the immediate asperities contact is separated by nanoparticles and the interaction between asperities and nanoparticles is negligible in ideal state, thus the shear stresses at the particle lubricating area are mainly friction force which are mainly determined by contact area between nanoparticle and a-C. As shown in Figure 11b,c, compared with NDs, the particle lubricating area of fullerenol dispersion is obviously higher, and thus having higher coefficient of friction. As reported by Cheng et al. [5], the larger size of particles, the less the particle numbers, and it is more easily tends to be pushed out the contact region, leading to a higher load carried by nanoparticles and non-uniform load distribution. However, the abrasive wear can be significantly reduced by fullerenol because of the lower load carried by fullerenol and more uniform load distribution.

The results of tribotest performed at different normal load conditions also furthermore confirms the above account. It is well known that the low load conditions are favorable for ball-bearing (rolling) action, and thus both fullerenol and NDs show good tribological properties at low load condition (see Figure 4). As the level of sliding friction increases with increasing contact load, the effects of fullerenol and NDs become less under high load conditions. Particularly for NDs, the adverse effects on sliding wear by particles occur, but the anti-friction effects are still apparent due to the reduction of the contact area.

## 5. Conclusions

Two types of rigid dot sizes carbon particles (i.e., fullerenol and NDs) are selected as lubricant additives to study dot size effects on tribological properties of a-C under water lubrication. Their lubrication effects at different additive concentrations are firstly comparatively investigated. The tribological tests demonstrate that fullerenol provides a better overall lubrication showing excellent effects in both friction and wear with a wider range of effective concentrations. However, it is hard for fullerenol to reach a friction coefficient as low as NDs. As no additive particles are observed on the worn surface and a-C film maintains the native structure during friction, it can be assumed that the main lubricating mechanism of fullerenol and NDs is the nano-bearing effect, and the nano size is the predominant factor causing the performance difference. For fullerenol, the nano-bearing effect takes place only at a concentration sufficient to support the asperity contact loads. Compared with NDs, the fullerenol nanoparticle lubricating area is higher and the load carried by particles is lower with more uniform load distribution in ideal state showing a better anti-wear effect. To further improve lubrication efficacy, the interactions between nano-bearing effect and interlayer shear sliding effect will be investigated in our future work.

## Figures and Tables

**Figure 1 nanomaterials-12-00139-f001:**
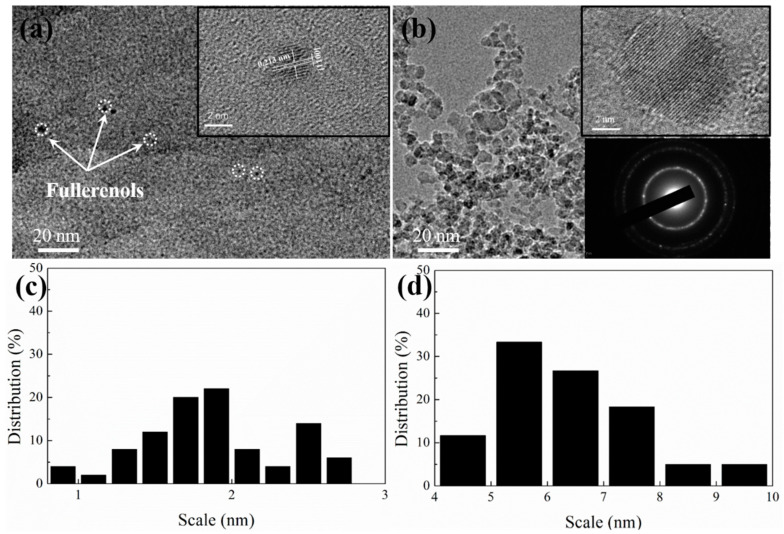
TEM image of (**a**) fullerenol and (**b**) NDs (inset is the HRTEM image); size distribution of (**c**) fullerenol and (**d**) NDs.

**Figure 2 nanomaterials-12-00139-f002:**
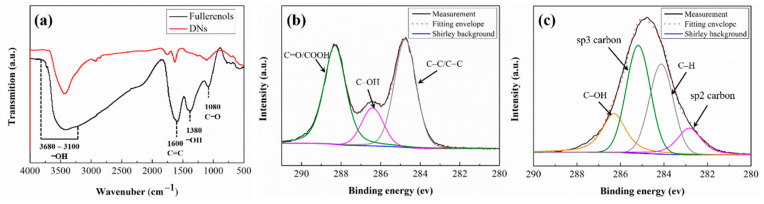
(**a**) FT-IR spectrum of fullerenol and NDs; C1s XPS spectra of (**b**) fullerenol and (**c**) NDs.

**Figure 3 nanomaterials-12-00139-f003:**
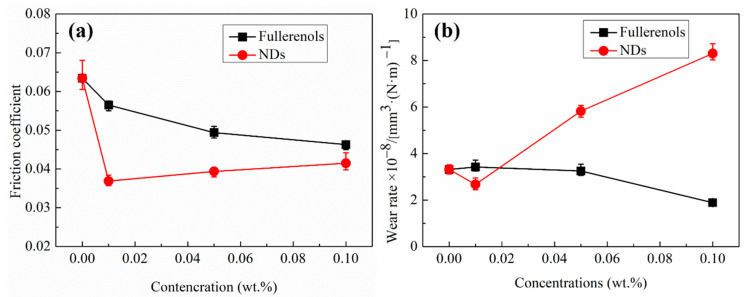
(**a**) Average COFs and (**b**) wear rates of a-C contacts lubricated with fullerenol dispersion and NDs dispersion with different additive concentrations.

**Figure 4 nanomaterials-12-00139-f004:**
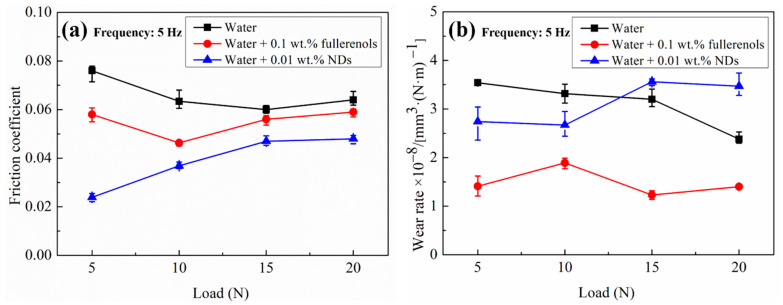
(**a**) Variation of COF with normal load and (**b**) variation of wear rate with normal load.

**Figure 5 nanomaterials-12-00139-f005:**
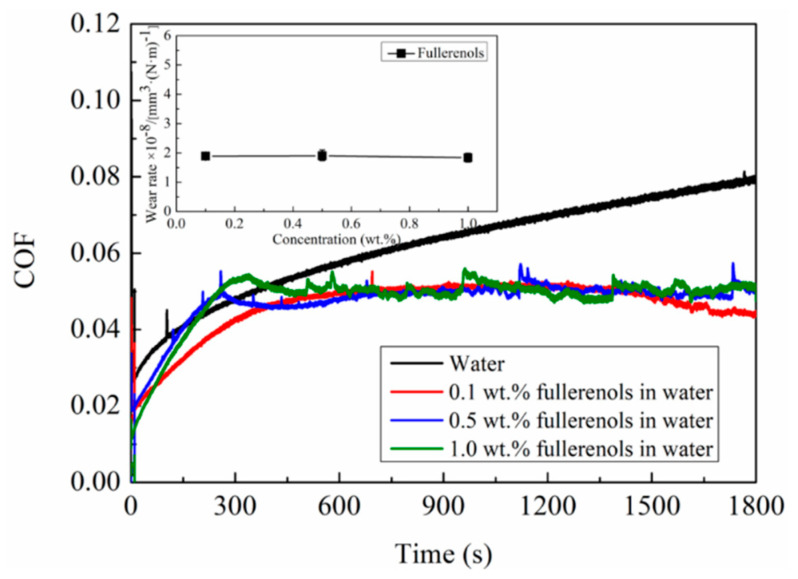
COF variation in a-C contacts as a function of sliding time when lubricants are water, 0.1 wt.%, 0.5 wt.%, and 1.0 wt.% fullerenol solution; inset is wear rates of a-C contacts.

**Figure 6 nanomaterials-12-00139-f006:**
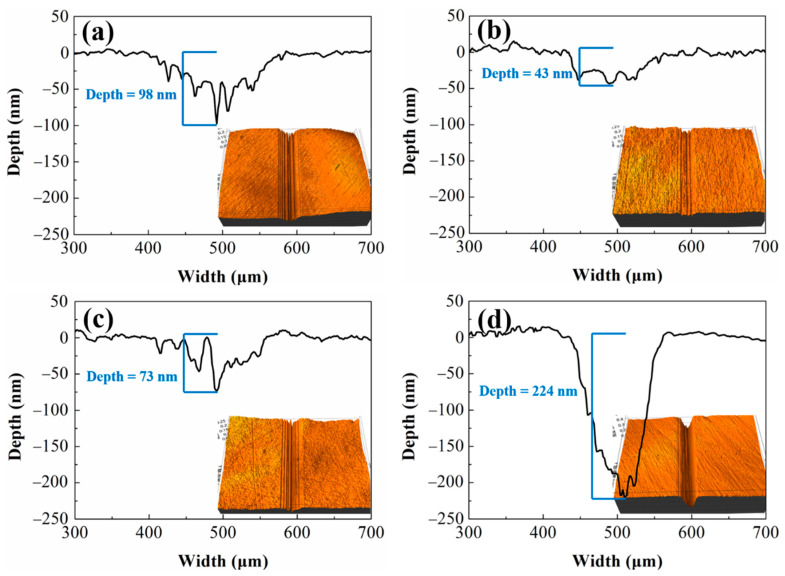
Noncontact 3D surface profiler images and profile curves of the wear tracks on a-C disks when lubricated with (**a**) deionized water, (**b**) water with 0.1 wt.% fullerenol, (**c**) water with 0.01 wt.% NDs and (**d**) water with 0.1 wt.% NDs respectively at a constant applied load of 10 N.

**Figure 7 nanomaterials-12-00139-f007:**
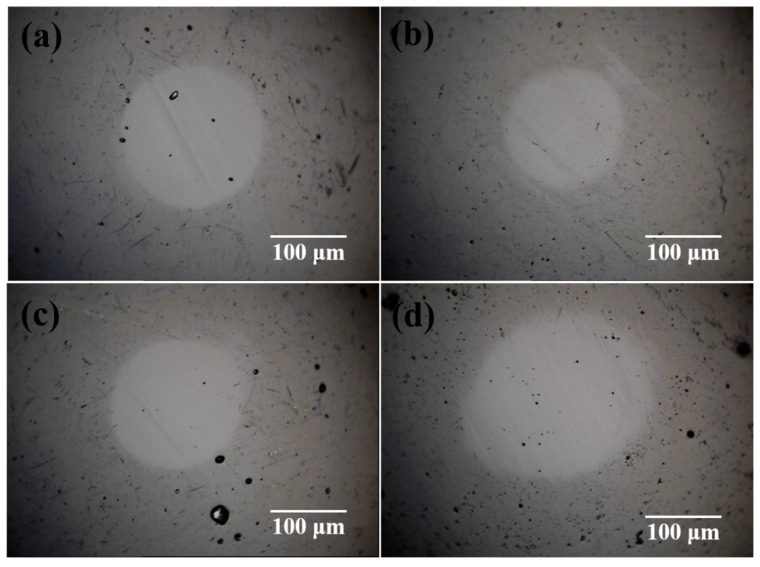
The optical micrographs of wear surfaces of a-C coated balls after sliding tests carried out under the applied load of 10 N in (**a**) deionized water, (**b**) water with 0.1 wt.% fullerenol, (**c**) water with 0.01 wt.% NDs, and (**d**) water with 0.1 wt.% NDs.

**Figure 8 nanomaterials-12-00139-f008:**
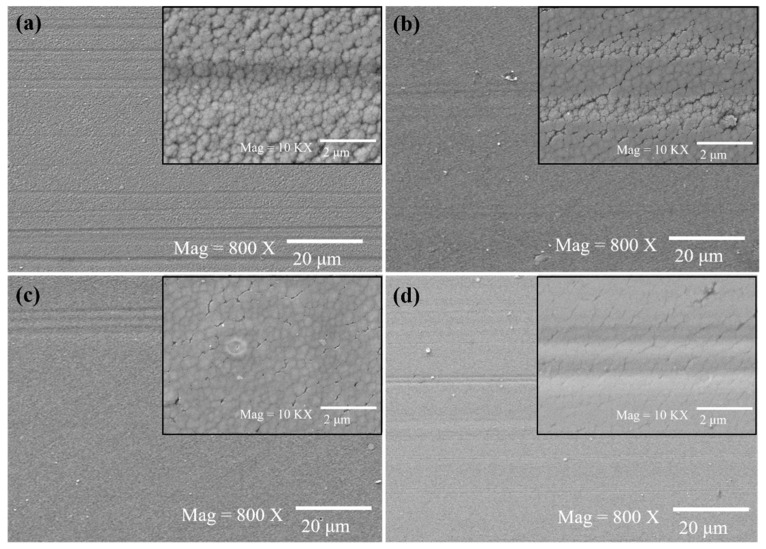
SEM images of wear tracks on a-C surface after sliding tests carried out under applied load of 10 N in (**a**) deionized water, (**b**) water with 0.1 wt.% fullerenol, (**c**) water with 0.01 wt.% NDs and (**d**) water with 0.1 wt.% NDs; insets are corresponding high-magnification SEM images.

**Figure 9 nanomaterials-12-00139-f009:**
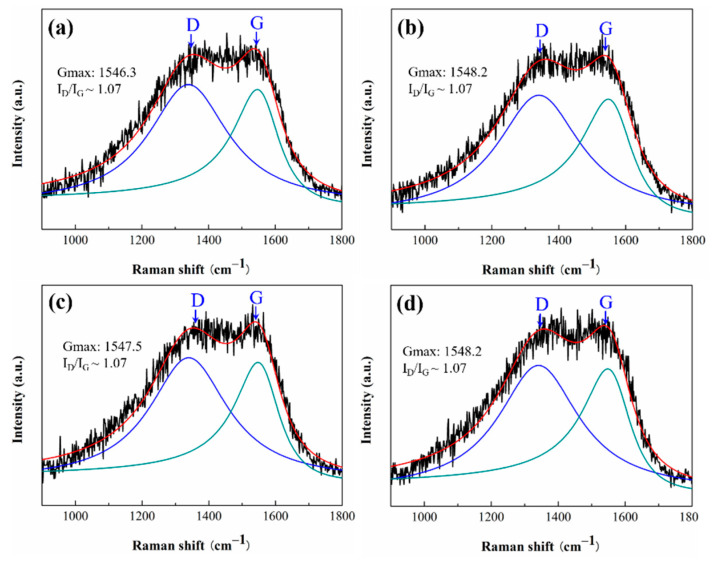
The Raman spectra of: (**a**) as-deposited a-C; and the wear tracks on a-C under the lubrication of: (**b**) deionized water, (**c**) water with 0.1 wt.% fullerenol, and (**d**) water with 0.01 wt.% NDs. The insets show the fitting results of G peak position (Gmax) and the I_D_/I_G_ ratio (height ratio).

**Figure 10 nanomaterials-12-00139-f010:**
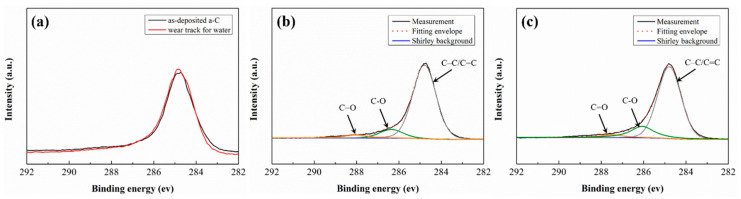
(**a**) C1s XPS spectra of as-deposited a-C films and wear tracks on a-C surface after sliding tests in deionized water; fitting results of C1s peak in XPS spectra of worn a-C surface under the lubrication of: (**b**) water with 0.1 wt.% fullerenol and (**c**) water with 0.01 wt.% NDs.

**Figure 11 nanomaterials-12-00139-f011:**
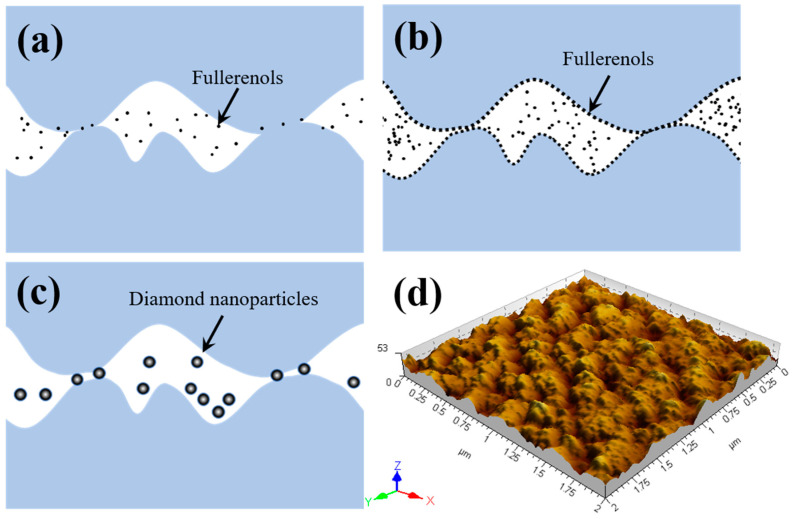
Schematic of tribological action mechanisms of (**a**) 0.01 wt.% fullerenol, (**b**) 0.1 wt.% fullerenol, and (**c**) NDs during tribological tests; (**d**) the topography of as-deposited a-C coating obtained by Atomic Force Microscopy.

**Table 1 nanomaterials-12-00139-t001:** The basic properties of as-deposited a-C coating.

Item	Properties
Coating method	Physical Vapor Deposition (PVD)
Carbon source	High-purity graphite
Transition layer	Cr interlayer
Thickness (μm)	3.0
Surface roughness, Ra (nm)	5.9 ± 0.6
Hardness (GPa)	10.6 ± 0.4
Young Modulus (GPa)	160.6 ± 5

## Data Availability

Data presented in this article is available on request from the corresponding author.

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
