# Peer review of "Study of Tribological Properties of Fullerenol and Nanodiamonds as Additives in Water-Based Lubricants for Amorphous Carbon (a-C) Coatings"

_nanomaterials, 2021, doi:10.3390/nano12010139_

Round 1

Reviewer 1 Report

In the paper, “Study of Tribological Properties of Fullerenol and Nanodiamonds as Additives in Water-based Lubricants for Amorphous Carbon (a-C) Coatings” lubricating properties of two different carbon nanophases are studied and really interesting results are obtained.

Different techniques have been used in order to carry out the characterization of the new lubricants, and their tribological performance. Authors also propose tribological mechanisms to explain their results

However, the manuscript contains several aspects that must be improved in order to take into account this work for publication.

-It is necessary to include the reference list.

-English spelling must be check, e.g. “nao” in line 18, “nanodimond” in line140, “performance tribological differences properties” in line 139…  

-It is necessary to include pictures of the different nano-lubricant in order to study their stability, because as it can be read in text, there are differences between nanodiamonds and fullerenol lubricants.

- It would be interesting to compare the tribological performance of the dispersions with the same concentration of nanophases in figure 4.

-I think GO (line 227) and XPS (line 290) have not been used in this study. Please check.

- I would like to know what load was used to obtain the wear tracks shown in Figure 6.

-It is not possible to see the scale/magnification used in SEM images (Fig. 7).

-It would be interesting to carry out a XPS analysis of dispersions and wear tracks.  

Author Response

Dear Reviewers:Thank you for considering the reviewing of my manuscript. I am grateful to be given a chance to revise the manuscript. I have considered the reviewer’s constructive comments carefully and a major revision is made.

In the revised version some important points have been rewritten and reorganized according to the comments. Some figures have been redrawn according to the modified explanations. We hope it will meet the requirement for publication. Revised portions in manuscript are marked up using the “Track Changes” function at the request of the editorial department. Here are the original comments and the main changes or explanations.

Reviewer 1:

In the paper, “Study of Tribological Properties of Fullerenol and Nanodiamonds as Additives in Water-based Lubricants for Amorphous Carbon (a-C) Coatings” lubricating properties of two different carbon nanophases are studied and really interesting results are obtained.

Different techniques have been used in order to carry out the characterization of the new lubricants, and their tribological performance. Authors also propose tribological mechanisms to explain their results

However, the manuscript contains several aspects that must be improved in order to take into account this work for publication.

-It is necessary to include the reference list.

Re: Thanks for the Reviewer’s suggestion. The references are listed at the end of original submission manuscript. The original submission may have been changed by editorial office that no references are shown in Manuscript for Revisions.

-English spelling must be check, e.g. “nao” in line 18, “nanodimond” in line140, “performance tribological differences properties” in line 139…

Re: Thanks for the Reviewer’s suggestion. The entire manuscript has been revised. If the English language still can’t meet the requirements for publication, we will use a professional English editing service recommended by the Editor before publication.

-It is necessary to include pictures of the different nano-lubricant in order to study their stability, because as it can be read in text, there are differences between nanodiamonds and fullerenol lubricants.

Re: Thanks for the Reviewer’s suggestion. Both fullerenol and nanodiamonds could be well dispersed in water and can remain suspended in water for 24 h, as shown in Fig. S1.

Corresponding changes in the revised version:(in line188-190 and Figure S1 in Supplementary Materials)

both fullerenols and nanodiamonds could be well dispersed in water and can remain suspended in water for 24 h (see Figure S1).

Figure S2. Digital images of dispersion of fullerenol and nanodiamonds in water with various concentrations (c, wt%). The time for each picture is noted on the respective picture.

- It would be interesting to compare the tribological performance of the dispersions with the same concentration of nanophases in figure 4.

Re: Thanks for the Reviewer’s suggestion. The tribological performance of the dispersions with the same concentration of nanophases are compared in the revised version and the corresponding results are shown in Figure S2 to explain the performance differences more concisely.

Figure S2. (a) variation of COF with normal load and (b) variation of wear rate with normal load.

Corresponding changes in the revised version: (in line249-254 and Figure S2 Supplementary Materials)

In addition, the tribological performance of the dispersions with the same concentration of nanophases are compared in figure S2. Just as in tribological results discussed above (Figure 3), at low concentrations of 0.01 wt.%, fullerenol have less effects on tribological performance of a-C, and at the concentration of 0.1 wt.%, nanodiamonds significantly worsen the anti-wear properties of a-C under water lubrication.

-I think GO (line 227) and XPS (line 290) have not been used in this study. Please check.

Re: Thanks for the Reviewer’s suggestion. We are very sorry for the negligence. The relative content has been revised and the XPS was measured via ESCALAB 250Xi XPS spectroscopy and added in the revised manuscript (see the response to the reviewer’s comments of XPS).

Corresponding changes in the revised version: (in line 248)

In addition, fullerenol dispersions exhibit much lower wear but higher friction compared with nanodiamonds dispersions under all condition.

- I would like to know what load was used to obtain the wear tracks shown in Figure 6.

Re: Thanks for the Reviewer’s suggestion. We are very sorry for the negligence. The load is 10 N, and information was added in the revised manuscript.

Corresponding changes in the revised version: (in line 269 and 295)

Figure 6 shows the morphology and cross-sectional profile of the wear track on the a-C surface after the tribological tests carried out under applied loads of 10 N.

Figure 6. Noncontact 3D surface profiler images and profile curves of the wear tracks on a-C disks when lubricated with (a) deionized water, (b) water with 0.1 wt.% fullerenol, (c) water with 0.01 wt.% nanodiamonds and (d) water with 0.1 wt.% nanodiamonds respectively at a constant applied load 10 N.

-It is not possible to see the scale/magnification used in SEM images (Fig. 7).

Re: Thanks for the Reviewer’s suggestion. We are very sorry for the negligence. In the revised version, the scale/magnification used in SEM images was reset to more direct and clear.

 Corresponding changes in the revised version: (in line 230)

Figure 8. SEM images of wear tracks on a-C surface after sliding tests carried out under applied loads of 10 N in (a) deionized water, (b) water with 0.1 wt.% fullerenol, (c) water with 0.01 wt.% nanodiamonds and (d) water with 0.1 wt.% nanodiamonds; insets are corresponding high-magnification SEM images.

-It would be interesting to carry out a XPS analysis of dispersions and wear tracks.

Re: Thanks for the Reviewer’s suggestion. the XPS of dispersions and wear tracks were measured via ESCALAB 250Xi XPS spectroscopy. The C1s peak of dispersions and wear tracks are automatically calculated by function and multi-peak assignment to derive the analytical peaks. The full width at half maximum of fitted peaks is set at 1.3 eV. In the revised version, the chemical composition of dispersions and wear tracks are discussed via XPS. The results also in the well agreement with the experimental results that we have already discussed in the original submission.

Corresponding changes in the revised version: (in line 191-198, line 342-348, Figure 2 in 204, Figure 10 in 349)

Figure 2. (a) FT-IR spectrum of fullerenol and nanodiamonds; C1s XPS spectra of (b) Fullerenol and (c) Nanodiamonds.

The chemical composition of fullerenol and nanodiamonds are further confirmed by XPS. As can be seen from Figure 2(b), the prepared fullerenol shows three main C1s peaks cen-tered at 284.8, 286.4 and 288.3 eV which are attributed to nonoxygenated carbon, monooxygenated carbon and hemiketal carbon respectively [33]. Four bands assigned for sp2 C-C bonds, sp3 C-C bonds, carbon bonded to oxygen as C-O and C−H bonds can be identified in the spectrum of nanodiamonds, just as in Figure 2(c) [34, 35]. In addition, the binding energies for fullerenol and nanodiamonds are obvious different because of the changes of the chemical state of carbon.

This result is also confirmed by XPS as no obvious changes in the C1s spectrums obtained from the a-C surfaces are found before and after tribological tests (see Figure 10). In addi-tion, the worn surfaces of a-C contains three types of C atoms: centered at 284.8 eV (C=C/C-C), 286.1 eV (C-OH) and 287.5 eV (C-O-C), which are the common chemical com-position of amorphous carbon film [36-38]. The characteristic peaks of as-prepared full-erenol and nanodiamonds are not found in the corresponding worn a-C surfaces, indi-cating there are no three body particles deposited on worn a-C surfaces.

Figure 10. (a) C1s XPS spectra of as-deposited a-C films and wear tracks on a-C surface after sliding tests in deionized water; fitting results of C1s peak in XPS spectra of worn a-C surface under the lubrication of: (b) water with 0.1 wt.% fullerenol and (c) water with 0.01 wt.% nanodiamonds.

Reviewer 2 Report

The topic developed in the present work is of current interest to the tribology community. The design of the experiment, the use of numerous spectroscopic and microscopy techniques to analyze the surfaces used and the presentation of the results are of a good standard. For these reasons we suggest the publication of this work after the correction/comment on the following details:

1) is the experiment conducted in full lubrication regime?

2) please use the same color for the same particles on figure 1 and 2

3) since particle dimensions play a major role in the results discusssion it would be helpful to show the diameter distribution of the nanodiamonds derived from the TEM images.

4) Please comment on wear contribution from the ball counterpart . 

5) XPS spectra are indicated in the main text but any XPS spectra is presented. Please give explanations

Finally, typesetting errors are present and english corrections are required on the whole text

Author Response

Dear Reviewers:

Thank you for considering the reviewing of my manuscript. I am grateful to be given a chance to revise the manuscript. I have considered the reviewer’s constructive comments carefully and a major revision is made.

In the revised version some important points have been rewritten and reorganized according to the comments. Some figures have been redrawn according to the modified explanations. We hope it will meet the requirement for publication. Revised portions in manuscript are marked up using the “Track Changes” function at the request of the editorial department. Here are the original comments and the main changes or explanations.  

Reviewer 2:

Comments and Suggestions for Authors

The topic developed in the present work is of current interest to the tribology community. The design of the experiment, the use of numerous spectroscopic and microscopy techniques to analyze the surfaces used and the presentation of the results are of a good standard. For these reasons we suggest the publication of this work after the correction/comment on the following details:

  • is the experiment conducted in full lubrication regime?

Re: Thanks for the Reviewer’s suggestion. The tribological test are carried out under boundary lubrication because of the very low viscosity of water. The theoretical minimum film thickness (hmin) and dimensionless lambda (λ) ratio can be calculated using Eq. (1) and Eq. (2). The calculated λ is lower than 0.002, suggesting that the lubrication is in boundary regime (λ<1).

  (1)

  (2)

where R’ is the reduced radius of curvature, U is the entraining surface velocity, W is the normal load, E’ is the reduced Young's modulus, ηo is the dynamic viscosity, α is the pressure–viscosity coefficient, Rq1 is the surface roughness of ball and Rq2 is the surface roughness of disc. The boundary lubrication regime was added in the revised manuscript.

Corresponding changes in the revised version: (in line 146)

The boundary lubrication friction tests are carried in ambient air (25% relative humidity) under 10 N applied loads with a frequency of 5 Hz (relative contact ve-locity of 0.05 m/s) for 30min.

  • please use the same color for the same particles on figure 1 and 2

Re: Thanks for the Reviewer’s suggestion. The setting of corresponding figure has been reset according to the reviewer’s suggestion.

  • since particle dimensions play a major role in the results discusssion it would be helpful to show the diameter distribution of the nanodiamonds derived from the TEM images.

Re: Thanks for the Reviewer’s suggestion. The statistical size distribution of fullerenols and nanodiamonds are measured from the TEM photographs using an image analysis software (nano measure 1.2). The results and corresponding discussion were added in the revised manuscript.

Corresponding changes in the revised version: (in line 175-179, and Figure 1 in line 200)

The statistical size distribution of fullerenols and nanodiamonds are measured from the TEM photographs using an image analysis software (nano measure 1.2). As can be seen from Figure 1(c), the fullerenols all have ultra-small diameters in the very narrow 1-3 nm range. The nanodiamonds have diameters of 4-10 nm and most particle diameter is larger than 5 nm.

Figure 1. TEM image of (a) Fullerenol and (b) Nanodiamonds (inset is the HRTEM image); size distribution of (c) Fullerenol and (d) Nanodiamonds.

  • Please comment on wear contribution from the ball counterpart.

Re: Thanks for the Reviewer’s suggestion. The images of wear scars on the slider balls and corresponding discussion was added in the revised manuscript.

Corresponding changes in the revised version: (in line 281-288, and Figure 7 in line 294)

The microscopic images of wear scar on the slider balls are shown in Figure 7. It is seen that compared with that of a-C lubricated by deionized water, the surface of wear scar is smoother and the diameter of wear scar is smaller under the lubrication of 0.1 wt.% fullerenol dispersion. This result suggests that introducing 0.1 wt.% fullerenol into water could alleviate the abrasive wear. Nanodiamonds show quite different effects. At the low additive amount of 0.01 wt.%, nanodiamonds had little effect on the appearance and diameter of wear scars but enlarge the diameter of wear scar and increase the number of deep grooves on wear scar at the additive amount of 0.1 wt.%.

Figure 7. The optical micrographs of wear surfaces of a-C coated balls after sliding tests carried out under applied loads of 10 N in (a) deionized water, (b) water with 0.1 wt.% fullerenol, (c) water with 0.01 wt.% nanodiamonds and (d) water with 0.1 wt.% nanodiamonds.

  • XPS spectra are indicated in the main text but any XPS spectra is presented. Please give explanations

Re: Thanks for the Reviewer’s suggestion. We are very sorry for the negligence. The results and corresponding discussion of the XPS was added in the revised manuscript. The XPS of dispersions and wear tracks were measured via ESCALAB 250Xi XPS spectroscopy. The C1s peak of dispersions and wear tracks are automatically calculated by function and multi-peak assignment to derive the analytical peaks. The full width at half maximum of fitted peaks is set at 1.3 eV. In the revised version, the chemical composition of dispersions and wear tracks are discussed via XPS. The results also in the well agreement with the experimental results that we have already discussed in the original submission.

Corresponding changes in the revised version: (in line 191-198, line 342-348, Figure 2 in 204, Figure 10 in 349)

Figure 2. (a) FT-IR spectrum of fullerenol and nanodiamonds; C1s XPS spectra of (b) Fullerenol and (c) Nanodiamonds.

The chemical composition of fullerenol and nanodiamonds are further confirmed by XPS. As can be seen from Figure 2(b), the prepared fullerenol shows three main C1s peaks centered at 284.8, 286.4 and 288.3 eV which are attributed to nonoxygenated carbon, monooxygenated carbon and hemiketal carbon respectively [33]. Four bands assigned for sp2 C-C bonds, sp3 C-C bonds, carbon bonded to oxygen as C-O and C−H bonds can be identified in the spectrum of nanodiamonds, just as in Figure 2(c) [34, 35]. In addition, the binding energies for fullerenol and nanodiamonds are obvious different because of the changes of the chemical state of carbon.

This result is also confirmed by XPS as no obvious changes in the C1s spectrums obtained from the a-C surfaces are found before and after tribological tests (see Figure 10). In addition, the worn surfaces of a-C contains three types of C atoms: centered at 284.8 eV (C=C/C-C), 286.1 eV (C-OH) and 287.5 eV (C-O-C), which are the common chemical com-position of amorphous carbon film [36-38]. The characteristic peaks of as-prepared fullerenol and nanodiamonds are not found in the corresponding worn a-C surfaces, indicating there are no three body particles deposited on worn a-C surfaces.

Figure 10. (a) C1s XPS spectra of as-deposited a-C films and wear tracks on a-C surface after sliding tests in deionized water; fitting results of C1s peak in XPS spectra of worn a-C surface under the lubrication of: (b) water with 0.1 wt.% fullerenol and (c) water with 0.01 wt.% nanodiamonds.

Finally, typesetting errors are present and english corrections are required on the whole text

Re: Thanks for the Reviewer’s suggestion. In the revised version, we have further polished the English writing throughout the manuscript. If the English language still can’t meet the requirements for publication, we will use a professional English editing service recommended by the Editor before publication.

Round 2

Reviewer 1 Report

Authors of “Study of Tribological properties of Fullerenol and Nanodiamonds in Additives in Water-based Lubricants for Amorphous Carbon (a-C) Coatings” have considered the suggestions provided and they have also answered kindly to the comments and doubts.

I have few comments to improve the quality of the paper:

-Please, review reference 16 (line 70), the first author is not Tang.

-Include reference 17 in line 73.

-Change reference 17 for Wang’s (reference 19) in line 74 and review reference 28, Konstantin is not the first author.

- There are some words separated by dashes in text, and they should not be there, e.g. Line 53, line 94, line 127, line 155, line 153, line 434…

- Reference 18 format must be reviewed.

- I would propose doing a frame surrounding the insets images in Fig 8 and Fig 1 to improve the understanding.

- It would be necessary to check some spellings mistakes as “pH” instead of “Ph”in line 136, “It is thereafter is” in line 129 or “with with” in line 166.

Author Response

Dear Reviewers:

  Thank you again for your valuable comments. It helps us a lot to improve our work. I am grateful to be given a chance to revise the manuscript. I have considered the reviewer’s constructive comments carefully and some revision is made. In the revised version, Fig 8 and Fig 1 have been redrawn according to the reviewer’s suggestions. Some mistakes of reference, spellings and words separated by dashes have been corrected. We hope it will meet the requirement for publication. Revised portions in manuscript are marked up using the “Track Changes” function at the request of the editorial department. Here are the original comments and the main changes or explanations.

Thank you for your consideration.

Sincerely Yours,

Jinzhu Tang

Corresponding authors:

Jinzhu Tang

E-mail: tangjinzhu11@163.com

Tel: +86-023-72791828

Addr. : College of Materials Science and Engineering, Yangtze Normal University, Juxian Rd 18, Chongqing, 408100, P. R. China.

Authors of “Study of Tribological properties of Fullerenol and Nanodiamonds in Additives in Water-based Lubricants for Amorphous Carbon (a-C) Coatings” have considered the suggestions provided and they have also answered kindly to the comments and doubts.

I have few comments to improve the quality of the paper:

-Please, review reference 16 (line 70), the first author is not Tang.

Re: Thanks for the Reviewer’s suggestion. I am sorry about this mistake. These writing errors have been corrected according to the reviewer’s suggestions.

Corresponding changes in the revised version: (in line 70)

For example, Liu et al. reported that the CDs-IL (ionic liquid) can increase the load-carrying capability and enhance the lubricity of water for steel contacts [16].

-Include reference 17 in line 73.

Re: Thanks for the Reviewer’s suggestion. I am sorry about this mistake. These writing errors have been corrected according to the reviewer’s suggestions.

Corresponding changes in the revised version: (in line73)

Xiao et al. found that sulfur-doped CDs show good anti-wear and anti-friction properties as water-based lubricant additives for Si3N4-vs-steel and Si3N4-vs-Si3N4 contacts [17].

-Change reference 17 for Wang’s (reference 19) in line 74 and review reference 28, Konstantin is not the first author.

Re: Thanks for the Reviewer’s suggestion. The reference for Wang’s in line 74 is reference 11 which have been corrected in the revised version. And the reference 28 is also revised accordingly.

Corresponding changes in the revised version: (in line 74 and line 125)

Hu et al. have researched the enhanced tribological properties and inhibition effect of CDs in water-based lubricants for 316 stainless steel [11].

The C60(OH)22–26 are synthesized by the most simple and repeatable method as pre-sented by Semenov K N [23] and Bityutskii N P [28].

- There are some words separated by dashes in text, and they should not be there, e.g. Line 53, line 94, line 127, line 155, line 153, line 434…

Re: Thanks for the Reviewer’s suggestion. I am sorry about this mistake. These writing errors have been corrected according to the reviewer’s suggestions. And we will continue to monitor this error.

Corresponding changes in the revised version: (in line 53 (water), line 94 (Additionally), line 128 (tetrabutylammonium), line 157 (tribological), line 155 (microscope), line 440 (sufficient), line 336 (additive particles), line 422 (low load), line 435 (concentrations), line 353 (indicating), line 354 and 303 (fullerenol), line 305 (nanodiamonds), line 135 (reprecipitation), line 103 (properties), line 69 (friction reduction), line 59 (lubrication))

- Reference 18 format must be reviewed.

Re: Thanks for the Reviewer’s suggestion. The reference 18 format have been reviewed in the revised version. In the revised version, all reference formatting has been updated according to ACS reference style as the format requirements of nanomaterials.

Corresponding changes in the revised version:

  1. Tang, J.; Chen, S.; Jia, Y.; Ma, Y.; Xie, H.; Quan, X.; Ding, Q. Carbon dots as an additive for improving performance in water-based lubricants for amorphous carbon (a-C) coatings. Carbon 2020, 156, 272-281.

- I would propose doing a frame surrounding the insets images in Fig 8 and Fig 1 to improve the understanding.

Re: Thanks for the Reviewer’s suggestion. In the revised version, a frame surrounding the insets images in Fig 8 and Fig 1 was done.

Corresponding changes in the revised version:

Figure 1. TEM image of (a) Fullerenol and (b) Nanodiamonds (inset is the HRTEM image); size distribution of (c) Fullerenol and (d) Nanodiamonds.

Figure 8. SEM images of wear tracks on a-C surface after sliding tests carried out under applied load of 10 N in (a) deionized water, (b) water with 0.1 wt.% fullerenol, (c) water with 0.01 wt.% nanodiamonds and (d) water with 0.1 wt.% nanodiamonds; insets are corresponding high-magnification SEM images.

- It would be necessary to check some spellings mistakes as “pH” instead of “Ph”in line 136, “It is thereafter is” in line 129 or “with with” in line 166.

Re: Thanks for the Reviewer’s suggestion. I am sorry about this mistake. These writing errors have been corrected according to the reviewer’s suggestions.
